# “Ko Au te Whenua, Ko te Whenua Ko Au: I Am the Land, and the Land Is Me”: Healer/Patient Views on the Role of Rongoā Māori (Traditional Māori Healing) in Healing the Land

**DOI:** 10.3390/ijerph19148547

**Published:** 2022-07-13

**Authors:** Glenis Mark, Amohia Boulton, Tanya Allport, Donna Kerridge, Gill Potaka-Osborne

**Affiliations:** Whakauae Research, Whanganui 4500, New Zealand; amohia@whakauae.co.nz (A.B.); tanya@whakauae.co.nz (T.A.); donna@oranewzealand.com (D.K.); gill@whakauae.co.nz (G.P.-O.)

**Keywords:** Rongoā Māori, traditional Māori healing, Māori, land, wellbeing, conservation

## Abstract

In Rongoā Māori (traditional Māori healing), the connection with the land stems from seeing Papatūānuku/Mother Earth as a part of our identity/whakapapa (genealogy), our culture, and our wellbeing. This qualitative study aimed to explore the holistic nature and meaning of Rongoā Māori. There were 49 practitioner and patient participants who participated in semi-structured interviews and focus groups across Aotearoa/New Zealand. The findings showed four themes: land as an intrinsic part of identity; land as a site and source of healing; reciprocity of the healing relationship; and the importance of kaitiakitanga/conservation to Rongoā Māori. Participants shared narratives of connections between the people and the land that showed that when the land is well, the people are well. Implications of these themes for Indigenous wellbeing and the conservation and protection of our natural environments led to three recommendations to reconnect with the land, support Rongoā Māori healing, and to participate in the conservation and preservation of local land and waterways. It is hoped that in learning more about the connection between the land and Rongoā Māori healing, we begin to place greater value on the need to conserve and preserve both the land and our connections to her through traditional healing practices.

## 1. Introduction

The connection between Rongoā Māori (traditional Māori healing) and the land is a story of the interconnectedness of the health and healing of both the people and the land. Māori, as the Indigenous people of Aotearoa/New Zealand, through the practice of traditional healing play an important care-taking and guardianship role with, and on behalf of, the land, one which provides mutual benefit, where Rongoā Māori heals both the people and the land. The purpose of this article is to demonstrate the critical need for re-establishing our connections to the land and the conservation and preservation of both the land and traditional healing practices as a means of ensuring health and wellbeing of people and planet. 

The article begins by outlining key concepts in te Ao Māori, the Māori worldview including connections between and importance of the land to Māori culture, identity, and wellbeing. Rongoā Māori, traditional Māori healing, is introduced and the relationship between this healing tradition and the land briefly outlined. The paper then introduces a research project that seeks to document the Rongoā Māori cultural interventions used by healers to restore and maintain health and wellbeing within their communities. Following a brief description of study methods, the article presents four key themes derived from research, discussing the implications of these themes for Indigenous wellbeing and the conservation and protection of our natural environments. 

The authors of this article are all Māori. All are university educated; four are trained and practicing researchers and one is a Rongoā practitioner, with many years of healing experience in the Rongoā sector. Individuals within the team are known within the Rongoā sector, either because of their previous research in the field or due to their being a practitioner themselves. The team is passionate about maintaining the cultural integrity of Rongoā Māori, and this research was born out of a desire to promote a greater understanding of Rongoā. The fact that our traditional healing practices are misunderstood, or not known at all, by many in Aotearoa/New Zealand has, in part, provided the impetus for this work. 

### 1.1. Te Ao Māori, the Māori World 

#### 1.1.1. Connection with the Land 

For Māori, the Indigenous people of Aotearoa/New Zealand, whenua (land), is a fundamental source of identity and spiritual connection and a significant environmental and cultural determinant of health. Māori regard themselves as ‘belonging to the land’, rather than ‘owning’ it because land is a tangible expression of whakapapa. One of the terms Māori use to identify themselves is tāngata whenua, literally “people of the land”, an identity derived from the belief that, as a people, Māori are one of the many offspring of a union between the Sky Father, Ranginui, and the Earth Mother, Papatūānuku [1]. 

For Māori, the relationship with the land shapes the ways in which the cultural, spiritual, emotional, physical and social wellbeing of people and communities is expressed. Māori believe that the starting point for health and wellbeing is a strong sense of interconnectedness with the landscape [2,3]. The environment is therapeutic, cultural and ancestral, where the caring of the landscape, and its conservation, is simply another form of mauri or life force of the people [3]. 

#### 1.1.2. Connection between Land and Māori Health and Wellbeing

Māori perceive natural ecological environments as a medium for physical, emotional, mental and spiritual health and wellbeing. Caring for land involves both environmental and resource management expertise and knowledge, based on long-held responsibilities and practices around the guardianship and protection of resources. Furthermore, caring for the land recognises the interconnectedness of the entire environment where the land becomes a part of the culture and identity of the people [3]. 

The importance of the land on the health of Māori relates to a sense of identity. As noted above, Māori have a strong relationship with the land, embodied as Papatūānuku, Mother Earth, shaping the way in which the people express their cultural, spiritual, emotional, physical and social wellbeing. Being born from the Earth establishes the foundation for the relationship between the individual, the whānau (family) and nature. The land becomes a medium for physical, emotional, mental and spiritual health and wellbeing through the whenua, te wao nui ā Tane (the forest), ngā wai ora (the water) and te rohe reporepo (wetlands) [4].

For Māori, access to natural places for health, spiritual well-being and cultural connections is vital. Contact with the land links Māori to mātauranga and tikanga (knowledge and customs), hinengaro and wairua (mind and spirit), tinana and tāngata (body and people) [5]. The practice of traditional healing (Rongoā Māori) includes the sustainable use of the whenua and moana (sea); the care, protection and appropriate use of mātauranga Māori (ancestral knowledge); and te reo Māori (the Māori language) [6].

A further study concluded that the principles of Māori healing include the mind, body, spirit, family and land as essential aspects of health and wellbeing. The interconnectedness of mind, body and spirit was highlighted, but the external relationships people have with their family/genealogy and with the land are viewed as just as important for maintaining good health as well as Māori customs, values and ways of life [7]. 

#### 1.1.3. Guardianship of the Land

The Māori worldview includes a complex system of practices and customs to ensure the sustainable use and care for environmental resources. In mātauranga Māori (Māori knowledge and knowledge systems), there is an emphasis on caring for the environment and of Māori being guardians, or kaitiaki, for the whenua; these practices are founded on deeply held spiritual beliefs [8].

Kaitiakitanga or guardianship is both a knowledge base and a set of practices. Kaitiakitanga enables Māori to maintain a relationship with the land, waters and natural resources, as well as retain an intimate knowledge of the events and relationships that have occurred in that area over time. The notion of kaitiakitanga involves more than simply interactions between people and the land to also embrace the concept of whakapapa (genealogy) and the application of whakapapa within the protection and management of ancestral lands [9]. Tino rangatiratanga (sovereignty) in relation to kaitiakitanga (guardianship) is concerned with the ability of local Māori to regulate relationships, between humans and the whenua. The rights of mana whenua (right of the people to manage the land) to act as environmental trustees and make decisions related to access and use of natural resources within the tribal territory must first be acknowledged. 

To support the role of kaitiakitanga, or guardianship, Māori follow a set of practices to govern resource use, known as tikanga, to ensure that the use of the land is both respectful to the ancestors and is sustainable. If the use of environmental resources is not sustainable, a rāhui (prohibition) can be placed by the local iwi (tribe) [8]. The critical role that Rongoā Māori plays in protecting and healing the land can be understood once the origins and knowledge base of this healing tradition are understood.

#### 1.1.4. Knowledge Base of Rongoā

Rongoā Māori derives from mātauranga Māori (Māori knowledge), which in turn forms the central pillar of a Māori worldview. Mātauranga Māori also informs the knowledge on traditional cosmology, where it is a part of humanity’s responsibilities as kaitiaki (care takers) to mediate between the needs of Papatūānuku and the atua (Māori gods). The role of humans is to express the consciousness of Papatūānuku and whakapapa together with Māori descent from the land; this relationship forms the fundamental ontological principles of Rongoā Māori. Māori have cosmological origins from Io Matua Kore, the supreme being, who created the universe and formed Papatūānuku and Ranginui (Sky Father). According to Māori cosmology, people are one of many “children” of Papatūānuku and Ranginui, others being Tāne Mahuta, god of the forests, or Tāwhirimātea, god of the winds, as well as all living inhabitants of the world—fish, plants, mammals and so on. The wairua, which is the underlying reality of this world, interconnects with the material world through mauri [10]. Māori relationships with the land extend beyond the physical to include cultural cosmic origins and narratives, as well as non-material understandings of the nature of this world.

Rongoā Māori (traditional Māori healing) includes the cultural concepts of and relationships with the environment, ecology, Māori cosmology, the atua (Māori gods), wairua, the ancestors, and tikanga (protocols) [11]. In essence, Rongoā Māori is about the practice of reciprocity with Papatūānuku, our Earth Mother [10]. Rongoā is a taonga tuku iho (ancestral treasure) of intergenerational healing of the people that becomes a connecting mechanism between the people and their cultural identity and connection to Māori customs and values, with conservation and protection of the whenua, as well as the power of the spiritual interconnection and the mauri of all life [6].

As a form of healing, Rongoā Māori has been used by Māori for centuries [12]. There are many healing modalities/practices), and some of the main elements of Rongoā Māori include the following: mirimiri and romiromi (body work/deep tissue massage); Rongoā rākau or wai rākau (plant medicines/herbal remedy); matakite (seer, gift of second sight, clairvoyance); karakia and wairua (prayer/spirituality) [11,12,13]. The properties of Rongoā healing extend beyond the physical and chemical properties of plant medicine to also encompass increasing the connection of mauri (life force) of person, plant and healer whereby this greater connection becomes the main goal of healing [11,12,14]. 

Although Rongoā has often been represented in the literature as no more than herbal medicines [15], this narrow understanding of Rongoā means that there are significant cultural elements to Māori healing that are overlooked. Rongoā is not only used to treat disease rather, the primary goal of Rongoā Māori is to restore and strengthen the mauri, the life force of the people and the land, and to do so requires a holistic perspective of both [6,11,16]. Rongoā is also regarded as a way of understanding the world and how to heal it, the interconnections between all things, and a way to strengthen connection to the whenua (land) through mutual respect between nature and man [11,14]. 

This article draws on a study that aims to document Rongoā Māori cultural interventions used by healers to restore and maintain health and wellbeing within communities. Through the course of the study, the components of Rongoā healing practice and philosophy in relation to people and the environment are explored in depth. The ultimate aim of the research project is to create a “biosphere”: a visual map of Rongoā as an integrated, holistic and dynamic system of healing relationships between the healer and the people and the spiritual dimension, plants, land and waterways. This article presents a subset of the data derived from that project, specifically the findings that explore the connection between the land and Rongoā Māori healing. 

## 2. Materials and Methods

This qualitative study used a mixture of Western and Māori methods of data collection, analysis and synthesis. Kaupapa Māori theory provides the theoretical framework for this research. The term Kaupapa Māori describes research that has been conceived, developed and carried out by Māori, with an end outcome that benefits Māori [17]. As a theoretical framework derived from a Māori worldview, Kaupapa Māori recognises Māori cultural values and systems, facilitates the use of Māori traditions and customs to be followed during research processes [17] and ensures cultural integrity of data is maintained during analysis [18]. In order to understand Māori issues, there must be a foundational base that has been built from Māori cultural values, including a respect for Papatūānuku rather than imported theories [18]. Kaupapa Māori research has been used in research as both a form of resistance to Western notions, as well as a methodological strategy. Employing a Kaupapa Māori theoretical framework for this study ensures that tino rangatiratanga (self-determination) remains a focus and outcome of this study [17,19]. 

In addition to adopting a Kaupapa Māori theoretical framework, the research project also adhered to the principles outlined in the Cultural, Ethical, Research, Legal and Scientific (CERLS) Guidelines to ensure research methods used would be culturally, methodologically and ethically appropriate for all parties involved [11]. The CERLS guidelines were the outcome of research that ascertained the Māori cultural principles that should be present in all Rongoā research. The CERLS guidelines advocated that in all Rongoā research, the cultural integrity of Rongoā is always maintained; full disclosure of all aspects of research is ensured; Rongoā healing principles are reflected in all methodology; the right of healers to hold and provide knowledge is maintained; and research methodologies used were culturally appropriate [11].

The data reported here represent the views of 49 healer and patient participants who were recruited from Aotearoa/New Zealand in the regions of Northland, Waikato, Manawatū and Hauraki. Demographic information about the participants is presented in Table 1 below. Purposeful sampling was employed to identify and select individuals or groups of individuals who would be able to provide a depth of information from their knowledge and experience about Rongoā Māori healing [20]. As noted above, this team has an in-depth knowledge of the Rongoā sector as a result of having worked in and with this sector for some years. As a team, and led by the knowledge of the team member who is a Rongoā practitioner, we collectively identified and recruited research participants by contacting individuals within the Rongoā sector by phone or email, and recruitment was conducted between March and August 2021. 

Participants ranged in age from 20 to 80 years. Some identified as a patient only, some as a healer and a patient and some identified as a healer only. Many participants explained that they had been both a patient and a healer in their healing experience. 

Once the healers and patients were recruited to the study, two meetings were held with each group or individual based on the preferences of each participant, and semi-structured interviews were conducted. Semi-structured interviews were conducted by various combinations of the authors guided by an interview schedule. Team members were trained in the use of the schedule prior to going out into the field. The face-to-face interviews were audio-recorded by the researchers and later transcribed by an external transcriber who was contracted for this purpose, who has signed a confidentiality agreement. After each interview, a summary of the interview was written up by team members as a “vignette” and these vignettes were returned to participants as a means of seeking validation of the information that had been captured. A koha (gift) of cash to recognise participant’s time and the knowledge that they had shared was provided to each participant. Furthermore, a koha of food was offered at the first interview/focus group. Ethical approval for the study was granted by the New Zealand Ethics Committee, Protocol NZEC20_36. 

Individual “case studies” for each participant or group of participants were collated from in-depth face-to-face interviews or focus groups [21]. The advantage of using a focus group is that they allowed for the generation of discussion among participants and a wider canvassing of ideas and theories [22]. Focus groups are a particularly useful technique for exploring cultural values and beliefs about health and disease, providing a safe discussion space to explore sensitive subjects [22]. 

Data collection occurred cyclically and recruitment by case study site was staggered depending on participant availability. Upon the completion of fieldwork in the case study sites, the research team began initial data analysis in order of interviews by employing the Rourou approach, a culturally appropriate Kaupapa Māori data analysis method [23]. Analysis ran concurrently alongside the case study recruitment and data collection. Each author conducted the interviews, and they were the only ones who completed data analysis: firstly individually and then together in a group discussion format. No software was used in the data analysis phase; rather, the research team coded and themed the data manually. Themes were tested and confirmed through a robust process of face-to-face deliberation and discussion as a collective. Final themes were arrived at upon consensus. 

Data reported in this article are preliminary findings that have been analysed according to the first two steps of the Rourou Māori approach [23], as discussed below. The Rourou Māori approach is based on the whakataukī (proverb) “Nāu te rourou, nāku te rourou, ka ora ai te iwi” which translates as “By your basket of food, and my basket of food, the people will be fed”. The whakataukī is traditionally used to signify the concept of manaakitanga (the care and feeding of the people). In a research context, we likened the food being gathered to the collection of knowledge, ideas and concepts. The whakataukī presents three steps that (1) acknowledges the kōrero (talk/stories) from each participant or focus group and then (2) considers the interpretation of the research team; finally, (3) it creates a collaborative and consolidated version of the analysis and interpretation. As the research is still ongoing, Step 3 is yet to be completed for the full dataset.

### 2.1. Case Study Analysis

Step 1 Nāu te rourou: Upon completion of each focus group/interview, the researchers for that interview created a vignette to summarise the input of the participants and sent to participants for feedback. 

Step 2 Nāku te rourou: In the second step, the research team followed a mahi-a-rōpū (collective group data analysis) method of data analysis and synthesis [23]. Researchers first completed their own individual analysis of each case, coding and theming the data, before coming together as a rōpū (group) at the data analysis hui to discuss the themes that had emerged from the analysis. At this stage, both intra-case analysis (i.e., what we have found at each of the case study sites) as well as deeper inter-case analysis occurred (i.e., comparing between cases). This process of data analysis was thorough and time-consuming, ensuring that the resultant themes captured the essential points made by each participant or group of participants.

Step 3 Ka ora ai te iwi: The final step, to be completed at a future date, will collate the data from across all case study interviews to create an overall analysis of all the case study data.

The preliminary data analysis findings presented in this article are provided from the interviews that had been completed and analysed to date (i.e., Steps 1 and 2 of the Rourou Māori approach to data analysis).

### 2.2. Limitations

The research was undertaken in four regions, all of which were in the North Island of Aotearoa/New Zealand. While participants came from a mix of urban and rural locations within those regions, a greater variety of locations, both urban and rural, may have allowed the research team to ascertain any differences in perceptions of land between those who live in the cities and those who live in the country. 

This study was conducted during COVID-19 lockdown periods in Aotearoa/New Zealand, and some interviews were conducted by phone or online Zoom calls. This may have impacted the interview process, especially as Māori are often non-verbal communicators. Had the interviews been conducted in person, different data may have been collected.

## 3. Results

Overall, the findings confirmed the close and interconnected relationship between people and the land. Rongoā Māori, the customs, therapies and modalities associated with traditional healing practice, was regarded by participants as a key component in the maintenance of that connection between Māori, as tāngata whenua and the land. In the case studies undertaken as part of a research project investigating the meaning of Rongoā Māori, four main themes relating to the connection between land and wellbeing were identified, namely: land as an intrinsic part of identity; land as a site and source of healing; reciprocity of the healing relationship; and the importance of kaitiakitanga/conservation to Rongoā Māori. 

### 3.1. Land as an Intrinsic Part of Identity

Participants in this study regarded the land as a part of their cultural identity. Participants spoke about the waters and the mountains of their local tribal region and their personal connection, through their whakapapa (genealogy), to those physical features of the land. Participants believed that the connection with the land derives from an intimate whakapapa connection—they are of the land and not separate from it.


*… that deep connection to whenua to moon to cycles to environment to everything that comes under that, and that’s how you become who are. And then you are of the land because you are of the land, you’re not above it, you’re with it. I think with that commonality in the past for some, it’s changed.*

*Key Informant, Hauraki*



*… the fundamental thing we have is we are earthbound people—we come out of the earth and we go back*

*to the earth, and unless we in actual fact acknowledge that connection we’re never going to be at peace. And so, to me the ultimate healing is reconnecting people back to whenua.*

*Focus group, Whanganui*



*What waters do I come from, and you come from? The waters of your mountains, but you also come from the waters of your mother and your waters of your mother go back to her mother, go back there, go back to Tāne, the first person, you know? So, our waters are a continuation of the start and so if we don’t take care of that we don’t take care of our mothers and we get sick.*

*Key Informant, Hauraki*



*For me it’s an energetic exchange between me and Tāne Mahuta or te moana, so if I think of them as being my tuakana [elder sibling] and my teina [younger sibling], they’ve got something for me and I’ve got something for them. Whether it’s something that I eat, something that I drink, or something that I put in to my skin or the intention of it through karakia or takutaku [chant], it’s a way of connecting myself to my bigger whānau, if that makes sense. It’s recognising that I’m a small … drop in the ocean or I’m the ocean in the drop, that it’s a reciprocal relationship between me and the taiao.*

*Focus group, Hamilton*


### 3.2. The Land as a Site and Source of Healing 

In addition to being an intrinsic part of their identity as Māori, participants viewed their connection to the land as a critical part of Rongoā Māori: that a close connection to the land was one way of maintaining overall health and wellbeing. Participants also observed that the natural environment was somewhere that was sought out for particular ailments and that the land itself has the power to heal.


*…I think Rongoā is a deliberate intentional act to connect me to the taiao, going to the māra kai [food garden] is an intentional act to grow food, to be part of that process, to get my hands dirty, and I don’t know the science about it, but I know I feel good when I’m in the māra kai. I feel good when I’m in the taiao. I’ve got a raru [issue], if I go to my moana and I have a bit of a pure [cleanse], I feel good, I feel restored. So it is a way of mitigating the mamae [pain] that you get being disconnected from the taiao in these concrete jungles and our little insular homes with our air conditioning and that splits us up from the cycle, the natural rhythms of life and I feel that the Rongoā gets me in to the ngahere [forest].*

*Focus group, Hamilton*



*… working with our people has been really fulfilling and there’s a lot of healing that goes on. There’s a lot of trauma that’s gone on and, you know, that trauma needs to be fixed. I think the land is one of the best healers out there.*

*Key Informant, Hauraki*


Participants shared the varied experiences they had of plant use in healing and the positive effects on themselves or their family. The land itself provided multiple plant and water mechanisms for healing the people.


*I actually use kawakawa [Piper excelsum] at home for our animals and find it really good. We use it ourselves as well and the kumarahou [Pomaderris kumeraho] as well … I use it in all the soaps that I make … I think people are really enjoying what we’ve got to offer there with the Rongoā.*

*Key Informant, Northland*



*… wai tote [salt water], it’s good for hakihaki [sores] … go in there, and they seem to, it washes away for some reason. The minerals that’s in the water, it cleans it up …*

*Focus group, Northland*



*…if we had hakihakis [sores] or, actually sometimes we didn’t even have hakihakis… if one of my cousins played up, she’d take us to the beach, you know? … so she always took us back to the water if someone was playing up or if someone was sick, if we had sores, and it was just natural, yeah, just like what we did, didn’t seem like anything strange … but cos my mum and that, part of the urban shift so that kind of knowledge, that connection was lost …*

*Focus group, Manawatu*


### 3.3. Reciprocity of the Healing Relationship

Participants in this study observed that in terms of healing, the land and the people have an interconnected, symbiotic and reciprocal relationship where the land has the capacity to heal the people, and the people have the capacity to heal the land. Participants expressed the view that the health of the land was reflected in the wellbeing of the people who inhabit or are from that land. Where there was trauma on the land, this was evident among the people of that place; the land and the people would mirror the health of each other. Therefore, if the land was regarded as “sick”, “unhealthy” or “unwell”, attention must be given not only to trying to heal the land but also to address the wellbeing of the people. This deep connectedness has, over time, also influenced how people identify with the land.


*We say “ko au te whenua, ko te whenua ko au; ko au te awa [river], ko te awa ko au” [I am the land and the land is me, I am the river and the river is me] but if our awa are sick and they’re being polluted…then who are we as people? I think you see a lot of that manifesting, you know, places with really sick land and their people are quite sick. The people had strong kaitiakitanga [guardianship roles] they often had strong land. That’s what I see anyway.*

*Key Informant, Hauraki*



*And the connection that people have…. there’s a lot of tears, people breaking down and crying, in a good way. But they’re actually feeling their maunga [mountain]; they’re feeling their waters and that’s part of them. But I think the more we know ourselves the stronger we are and the stronger we are, you know, it’s like a snowball effect.*

*Key Informant, Hauraki*



*… before you can heal anybody you have to heal the land itself. So, you know, it’s looking at, again looking at the environment and, and finding out, is there gonna be enough resources for this Rongoā that everyone is gonna start making for the people? So that’s really hit home for me, is let’s look at the land first, so, yeah.*

*Focus group, Northland*



*You can also tell … what problems are in a particular area just by looking at the environment. You can straight away know what the health issues are gonna be, you know, just by looking at what they’re ingesting, what people are ingesting in that area.*

*Focus group, Northland*


People also shared experiences where they felt there was a significant disconnect between people and the land which in turn created ill-health, feelings of despondency and a sense of powerlessness.


*I can hear the earth crying. The crying is so persistent and so strong that I can’t, in actual fact, walk away from it even if I tried to at times, but I feel that the sad thing is that most people can hear the crying, but they don’t know what it is.*

*Focus group, Whanganui*



*I think from back in places there was a lot of trauma on the land. There’s a lot of trauma on the land and that sometimes will change the mauri of the place. And connecting people back I feel like our tupunas are always still there, they’re waiting for us, and we don’t go visit anymore. We don’t talk to the trees anymore. We don’t talk to our tupunas anymore.*

*Key Informant, Hauraki*



*… the most fundamental cause of ill health is loss of connectedness to whenua.*

*Focus group, Whanganui*



*Look at how we’ve treated Papatūānuku. We’ve abused her. We’ve misused her. We’ve taken and taken and taken, and somehow, we live our life with a guilty conscience and that is a source of sickness. So, I think fundamentally, earth in general, earth as mother. That’s the connection that we lack because if a man or a woman, or anyone grows up without their mother’s love have never felt that warmth.*

*Focus group, Whanganui*



*The loss of the ability to act as voices for the whenua. The powerlessness that people feel and in actual fact transfers into an inability even to care for themselves.*

*Focus group, Whanganui*


### 3.4. The Importance of Kaitiakitanga/Conservation to Rongoā Māori 

Participants believed that kaitiakitanga or the conservation and protection of the land was a part of the cultural role and responsibility of Māori. However, for a range of reasons, not the least being a history of colonisation, assimilation and land loss, participants observed that many Māori today are disconnected from their historical stories and traditional lands. Being disconnected from culture, history and traditions negatively impacts the ability of individuals and communities to have a relationship with the land, let alone a relationship that protects or safeguards the land.


*I spent a lot of time outdoors when I was young. So that was quite a natural thing and part of that, the kaitiakitanga that comes along, I’m really realising now that a lot of our youth have never seen it or got the opportunity to see it and therefore that’s an area that I’m quite passionate about*

*Key informant, Hauraki*



*The stories haven’t been passed on and in actual fact, when the stories fade away, to me that’s [a huge loss]. Māori are finding their connections are strained. How can [you] be a kaitiaki when you don’t know what you’re looking after?*

*Focus Group, Whanganui*



*The loss of the ability to act as voices for the whenua. The powerlessness that people feel and in actual fact transfers into an inability even to care for themselves.*

*Focus group, Whanganui*


Participants observed that where the people have been disconnected from their local environmental knowledge or from having the ability to participate in taking care of the local environment, they experience profound emotional and spiritual disconnection. A sense of responsibility to care for the land is part of the Māori cultural responsibility, and when that responsibility is unable to happen, it can affect the health and wellbeing of the people. However, participants also shared the ways in which Māori have been able to advocate, to re-learn about the cultural significance of land, to become involved in the conservation of the natural environment and to have a voice.


*… our wetlands are trying to talk to us and we are not listening to them, and I think we romanticise ourselves a little bit. I’m a kaitiaki. …And so, the space I’m currently in is about trying to get that conversation happening again, between our people and their whenua. Well, mostly in that repo [wetlands] space, and then allowing that combined narrative to then be shared with the rest of the world… for a very long time apparently Māori had nothing to do with wetlands… [but it] was really important to say, “Well, actually we have something to offer, and there’s a lot you can learn from our people in that regards.”*

*Focus group, Whanganui*


Participants also shared how work conserving birdlife and their environments also extended to the conservation of surrounding habitats and the impacts those habitats in turn have on the birdlife.


*If you’re talking about Rongoā, a lot of our people have taken the thing of clearing out exotic trees and bringing back Indigenous trees because its Rongoā for them. Forgetting that we have a connection of whakapapa to all these other things that have also been just as colonised as us and so the story that we were going out about the ruru [owl], was that our ruru have had to adapt to the same things that we’ve had to and if we’ve going to try and make the world a better place for us we’ve got to remember… those animals, those species, those plants, whatever…. We’ve got to think about what they need too. And yeah, we’ve heard that the cause of the ruru have dropping as the big trees are being removed for river restoration. So, they’re taking down pines, like grandaddy pines, gums, poplars. If you find a bat in there, they’ll protect the tree. If a ruru is using it, they don’t care. So, we were trying to get that message out that the ruru are using them as well.*

*Focus Group, Whanganui*


## 4. Discussion

The findings have shown that the participants identify strongly with the land as a part of their cultural identity, that land is both a site and a source of healing, that the land and people share a reciprocity of health and healing and that kaitiakitanga or conservation of the land is vital to the health of Māori. In this section, the implications of these themes for Indigenous wellbeing and the conservation and protection of our natural environments are discussed.

Māori, in common with other Indigenous peoples around the world identify strongly with and see themselves connected to the land, place or country from which they hail. The interconnection between Māori as an Indigenous people and the land that they occupy and inhabit has been well documented in the academic literature [2,4,5,6,7,8,10,11,24,25,26]. For Māori, that relationship is one based on a whakapapa or genealogical connection [1,5,10,27]. This study supports the understanding that, as an Indigenous people, the health and wellbeing of Māori is intrinsically connected to being able to access traditional lands and maintain a living, reciprocal and harmonious relationship with those lands. Irrespective of upbringing or current place of residence, participants in this study confirmed that the land is part of their cultural identity. In addition, participants spoke about the importance of the land as a source of wellbeing—physically, emotionally and spiritually. 

In cases where Māori are disconnected from their traditional lands, their tūrangawaewae, the likelihood of poor physical, emotional, mental and spiritual health is increased [3]. Participants observed feelings of despondency, desolation and mental anguish as being the consequence of not having a connection to or relationship with their traditional homelands. A lived connection to and understanding of one’s land can support the flourishing of one’s mauri (vitality); conversely, a lack of connection may lead to the state which Durie [28] has termed mauri noho (languishing) or mauri rere (unsettled). 

Traditional healing practices are also dependent upon and intimately tied to the land. More than simply a source of the raw plant material required for many of the herbal remedies created, the land has a specific whakapapa relationship to Rongoā Māori. Wikaire [16] describes Rongoā as involving a deep understanding of the environment as well as the narratives associated with local surroundings. The practice of Rongoā Māori involves a connection to and interaction with the natural environment such as the water, sun and wind, all of which are essential to healing practice [16]. Rongoā Māori reflects a working knowledge about the land that holds human origins and embraces environmental and celestial interconnections and spiritual communication with the plants and trees. 

Participants in this study observed that whereas people and land were reliant on each other for their wellbeing, so too Rongoā Māori and the land are reliant on each other for their respective wellbeing. Where the land or the water is despoiled, polluted, contaminated or diseased, Rongoā Māori medicine plants may be absent or less effective than those gathered from flourishing and healthy lands and waterways. Equally, however, if the land or waterway is treated with Rongoā Māori, places of despoilation may be restored and rebalanced by applying the principles of Rongoā to land management [29]. Studies show that kaitiakitanga takes place through connections to local whānau, hapū (sub-tribe) and iwi tribal regions, as well as the connection to Papatūānuku and the ancestors and kin. There needs to be engagement between the individual and the environment, strengthened through intergenerational knowledge sharing and spirituality because the mauri of the environment and the people are interconnected [30]. The findings from this study reveal that the wellbeing of both humankind and the land alike can be moderated and facilitated through the practice of Rongoā Māori.

Rongoā Māori plays a key role in supporting efforts to rebalance or regain health of our lands and waterways. As a consequence, the importance of preserving and conserving the land is directly interconnected with the conservation and preservation of Rongoā Māori, and vice versa. The application of Rongoā Māori practices, techniques and therapies to the healing and restoration of the land ultimately serves to benefit Māori, given the deep connection between Māori and the land. What this study offers, therefore, is not only additional evidence of the interconnectedness between Māori and the land, but more importantly, the critical role that Rongoā Māori plays in maintaining and, where necessary, restoring the health, wellbeing and vitality of the people and land.

### Recommendations

Māori identification with the land reinforces cultural and social connection. As alienation from ancestral land reduces Māori health and wellbeing, the land may also be seen as a reverse pathway to Māori empowerment [24]. The findings of this study show both the value and need for conservation and preservation of both the land and traditional healing practices because they each ensure the health and wellbeing of the other. Three recommendations are proposed as a means of maintaining a connection between Māori and the land, namely that Māori individuals and collectives:Take active steps to reconnect with their lands through interaction with and care of the land as pathways towards wellbeing;Support Rongoā Māori healing for the health and wellbeing of both the people and the land;Take active steps in the conservation and preservation of local land and waterways.

These three recommendations will help reconnect the people with the land and ensure that the mutual health and healing of both the land and the people remains sustainable. Māori could interact with the land by growing home gardens, or participating in marae (tribal meeting house complex) or community gardens. Land-based healing practices can take place, such as harvesting, education, ceremony, recreation and cultural-based counselling [31]. Practicing or learning about Rongoā Māori would mean a greater engagement with the land. Providing opportunities to learn about and practice Rongoā may be encouraged as way of re-establishing Māori connections to the land and the culture that has grown from it. Learning about the depth and breadth of Rongoā Māori would a help protect the practice of Rongoā from modern tendencies to reframe the practice as physical medicine for the human race only. Advocating that Māori take part in environmental land activities, such as protecting local wetlands or birds, may be a way to reconnect both the people and the land. This would provide a way to appreciate the significance of land-based practices as a catalyst for regenerating Indigenous social, spiritual and physical land-connection [32].

The implications of these recommendations for the Rongoā sector could be a renewed focus on both Rongoā healing and the land simultaneously. It may be the responsibility of iwi leaders and organisations to ensure that Rongoā healing and access to the whenua are a strategic part of health initiatives and programs for future consideration. Currently, Māori health initiatives are very focused on individual and whānau health participation with little mention of healing or accessing the whenua [33]. This study would advocate for a renewed focus on learning Rongoā healing principles and accessing tribal lands as an integral part of Māori health strategy. In addition, given the climate crisis that is affecting not only Indigenous peoples but humanity in general [34], further research on the role Rongoā Māori might play in mitigating the effects of climate change and maintaining the health and wellbeing of the people and the land in future may be of immediate value.

## 5. Conclusions

The aim of this article was to explore the connection between the land and Rongoā Māori healing in order to demonstrate how the conservation and preservation of both the land and traditional healing practices supports the health and wellbeing of people and planet. This study has demonstrated that an Indigenous connection with the land involves knowledge about that land and its history, as well as working on and for that land. This creates mutually beneficial health and wellbeing, and Rongoā Māori is a healing mechanism between the two. 

It is advocated that Māori reconnect to the land, support Rongoā Māori healing practices and actively participate in the conservation and preservation of local land. Each of these actions would contribute to a continued reconnection between the land and the people.

The conservation and preservation of both the land and traditional healing practices is a key means by which Indigenous people can support their wellbeing and the wellbeing and vitality of the land which they inhabit. In a modern world where connection to the land is becoming less important, this is a vital reminder that our future may not lie in technology, but in taking care of the natural world around us to ensure our own survival.

## Figures and Tables

**Table 1 ijerph-19-08547-t001:** Participant demographics: this table presents the participant demographic information by gender, age, ethnicity and occupation.

	Focus Groups and Interviews
Gender	Female: 34 Male: 13 Did not complete: 2
Age range	21–80 years
Ethnicity	37 Māori 8 European 1 Pacific Did not complete: 3
Occupation	Health worker (social work, nurse, manager, mental health, occupational therapist, tikanga and cultural facilitator, community worker, kaimahi (worker), disability worker, health and safety rep, physiotherapist, kaumatua (elder), health promotion): 15 Student: 6 Other (lead systems innovator, independent researcher, small business owner, managing director, kaiako (teacher), revenue and reservations manager): 6 Elder, koroua (elderly male), retired: 4 Mother: 3 Rongoā practitioner: 3 Natural therapies practitioner: 1 Did not complete: 11

## Data Availability

The data for this study is currently securely held by Whakauae Research Services Ltd., at their premises in Whanganui, New Zealand. Individual participant data will not be available. This was a requirement of the ethics approval for confidentiality of information NZEC20-36. More detailed information regarding methodology and interpretation of de-identified, aggregated data are available from the authors on reasonable request.

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
