# Peer review of "“Ko Au te Whenua, Ko te Whenua Ko Au: I Am the Land, and the Land Is Me”: Healer/Patient Views on the Role of Rongoā Māori (Traditional Māori Healing) in Healing the Land"

_ijerph, 2022, doi:10.3390/ijerph19148547_

Round 1
Reviewer 1 Report
The difficulties encountered in describing and explaining the values and knowledge system of Māori in the language of the British invader and coloniser to a non-Māori audience cannot be underestimated. The authors have carefully and patiently explained the inseverable links between Māori and all aspects of the natural world. The links are so deeply embedded that even though British attempts to destroy them visited incalculable damage to Māori, they were preserved by those specially trained to do so.
This paper draws on the knowledge and expertise of a selection of those specialists to provide explanations that are clear and simple, and make basic common sense to Māori who still know that world. It is through specialists such as these that the tikanga (law) that regulates the Māori world is gradually being restored, reinvigorated, and strengthened so that the necessary healing that rongoā Māori provides can take place - for both the people and their natural world relatives.
This paper provides a well written, carefully described and logically argued explanation for why and how this is happening. It is most refreshing to see our thinking and world explained so well. I found it a pleasure to read.
Author Response
Our team would like to extend our thanks to the reviewers for the time they took to review this paper. We appreciate their positive comments and valuable feedback. We purposely highlighted the links between Māori and the natural world as an intrinsic part of our cultural identity in a way that would be understandable for those who are not familiar with Māori culture. It is very important to us to preserve and maintain the cultural integrity of Rongoā Māori healing practices and we hope this article will highlight that.
Thank you for your review and for acknowledging the link between Māori and all aspects of the natural world. We did aim to make our explanations understandable for all and we do believe that Rongoā has the ability to heal both the people and the land. Thank you for taking the time to make comment and we note that there are no specific comments to address.
Reviewer 2 Report
Please find attached my comments to the authors and editors.

Author Response
Our team would like to extend our thanks to the reviewers for the time they took to review this paper. We appreciate their positive comments and valuable feedback. We purposely highlighted the links between Māori and the natural world as an intrinsic part of our cultural identity in a way that would be understandable for those who are not familiar with Māori culture. It is very important to us to preserve and maintain the cultural integrity of Rongoā Māori healing practices and we hope this article will highlight that.
Thank you for your valuable comments which will help us to improve the article.
Introduction
Reviewer 2 has suggested that the introduction to this article should be shortened. However as Reviewer 1 has observed we have taken great care to explain Māori values and knowledge system to a non-Māori audience which cannot be underestimated. We have made some edits in the introduction section to try and reduce its length, without deleting key concepts which must be understood if the remainder of the paper and indeed the research findings themselves are to be understood.
Methods
We have provided the following answers to your questions
- How was the sampling technique determined (rationale)? Please include an explanation.
Purposeful sampling (Palinkas et al, 2015) was chosen for this project as we specifically required key informants who not only had a deep understanding of Rongoā Māori but were also willing and able to articulate that knowledge and experience regarding Rongoā Māori.
We have modified the article in L235 as followed:
Participants were recruited using purposeful sampling to identify and select individuals or groups of individuals who would be able to provide a depth of information from their knowledge and experience about Rongoā Māori healing (Palinkas et al, 2015).
- What is the researcher’s indigeneity (if any)?
The authors are all Māori women and one of our team is herself a Rongoā Practitioner. We have included an author positioning paragraph on L48 to provide further explanation, which is at the request of another reviewer. In addition, the bios of all authors will be included with the article for the information of readers. The additional paragraph is as follows:
The authors of this article are all Māori. All are university educated; four are trained and practising researchers and one is a Rongoā practitioner, with many years of healing experience in the Rongoā sector. Individuals within the team are known within the Rongoā sector, either because of their previous research in the field, or due to their being a practitioner themselves. The team is passionate about maintaining the cultural integrity of Rongoā Māori, and this research was born out of a desire to promote a greater understanding of Rongoā. The fact that our traditional healing practices are misunderstood, or not known at all, by many in Aotearoa/New Zealand has, in part provied the impetus for this work.
- Clarify who transcribed the interviews and if they were sent back to participants for validation and accuracy check. Clarify if participants provided feedback on the findings and how it happened.
We have provided further explanation at L261 as follows:
The face-to-face interviews were audio-recorded by the researchers and later transcribed by an external transcriber who was contracted for this purpose, who has signed a confidentiality agreement.
We have provided further explanation at L263 as follows:
After each interview, a summary of the interview was written up by team members as a “vignette” and these vignettes returned to participants a means of seeking validation of the information that had been captured.
- Flush more details on how the participants were recruited (the sampling technique is mentioned, but not the recruitment and engagement strategy).
We have provided further explanation at L239 as follows:
As a team, and led by the knowledge of the team member who is a Rongoā practitioner, we collectively identified and recruited research participants by contacting individuals within the Rongoā sector by phone or email and recruitment was conducted between March – August 2021.
- Were interviewers trained to perform the data collection? What measures were taken to control (and assure) consistency between and within qualitative interviews?
We have provided further explanation at L256 as follows:
Team members were had been trained in the use of the schedule prior to going out into the field.
- State number of interviewers and data coders, as well as their initials (if they are authors of the paper).
We have provided further explanation at L283 as follows:
Each of the authors both conducted the interviews, and were the only ones who completed the data analysis, firstly individually, and then together in a group discussion format. No software was used in the data analysis phase, rather the research team coded and themed the data manually. Themes were tested and confirmed through a robust process of face-to-face deliberation and discussion as a collective. Final themes were arrived at upon consensus.
- Clarify and identify if any software (eg NVIVO) was used during the analysis.
As we did not use any software to assist us code the data, we have provided further explanation in the paragraph as noted above at L283.
- Discuss data saturation in relation to a large sample size.
As we have now clarified in the article, we have not yet completed our interviews for the study and have yet to determine if we have reached a point of data saturation. This article reports our preliminary findings from the study, and we are currently conducting the remaining outstanding interviews. Once these data have been collated, we will have reached a point of data saturation and will be reporting on total findings. In our view our early findings from our study regarding the links between wellbeing and land were sufficient to prompt the writing of this article.
- How many data coders coded the data?
There were 5 data coders who coded the data, which are the 5 authors of this paper. We included further explanation about this in the paragraph at L283 which is listed above.
Results and Discussion
- Please include dates defining the periods of recruitment
We have included the following sentence at L243 as follows:
… and recruitment was conducted between March – August 2021.
- I’d suggest including a table showing participants demographic
A table presenting the participant demographic has been included at L249 as follows:
Table 1: Participant demographics: this table presents the participant demographic information by gender, age, ethnicity and occupation.
|
|
Focus groups and interviews |
|
Gender |
Female: 34 Male: 13 Did not complete: 2 |
|
Age range |
21-80 years |
|
Ethnicity |
37 Maori 8 European 1 Pacific Did not complete: 3 |
|
Occupation |
Health worker (social work, nurse, manager, mental health, occupational therapist, tikanga and cultural facilitator, community worker, kaimahi (worker), disability worker, health and safety rep, physiotherapist, kaumatua (elder), health promotion): 15 Student: 6 Other (lead systems innovator, independent researcher, small business owner, managing director, kaiako (teacher), revenue and reservations manager): 6 Elder, koroua (elderly male), retired: 4 Mother: 3 Rongoā practitioner: 3 Natural therapies practitioner: 1 Did not complete: 11 |
Please provide a description of the coding tree (if any).
Instead of a coding tree, we held analysis discussion meetings, as further explained in L283 as above.
Conclusion
- Maybe some other implications for future research should be added.
In alignment with your suggestion below, we have added that future research could be conducted on the perspectives of the Rongoā Māori sector on climate change at L646 as follows:
In addition, and given the climate crisis that is facing not only Indigenous peoples but humanity in general [36], further research on the role Rongoā Māori might play in mitigating the effects of climate change and maintaining the health and wellbeing of the people and the land in future may be of immediate value.
- I would also encourage adding to the conclusion or discussion section reflections on how the health impacts of climate change on Māori people are interconnected and far-reaching.
While we recognise this is a significant issue for Indigenous people including for Māori of Aotearoa, the issue of climate change was not the focus of this study, nor is it the expertise of our team. We recognise that “the health impacts of climate change on Māori people are interconnected and far-reaching and result from direct and indirect impacts of climate change that exacerbate existing inequities and affect food and water security, air quality, infrastructure, personal safety, mental health, livelihoods, and identity, as well as increase exposure to organisms causing disease.” However, asking the team to write about the relationship between “climate change interlinked with the 17 SDGs and the impact on Māori people’s health” is outside the scope of this article and was not our purpose in writing it.
In recognition of the reviewer’s interest in this area, we have added climate change as a possible topic for future research within the Rongoā sector. It may well be, if our data supports such a piece of writing in the future that we will look to exploring these issues in a subsequent paper. However, for this particular article we humbly decline the suggestion to insert new material in either the discussion or the conclusion regarding the “health impacts of climate change on Māori people”. If the Reviewer is interested in research in this specific area, we would refer them to the work that is being done in this field particularly by Jones and colleagues:
Jones R. Climate change and Indigenous Health Promotion. Global Health Promotion. 2019;26 (3_suppl):73-81. doi:10.1177/1757975919829713
Jones, R., Bennett, H., Keating, G. and Blaiklock, A., 2014. Climate change and the right to health for Maori in Aotearoa/New Zealand. Health & Hum. Rts. J., 16, p.54.
Bennett, H., Macmillan, A., Jones, R., Blaiklock, A. and McMillan, J., 2020. Should health professionals participate in civil disobedience in response to the climate change health emergency?. The Lancet, 395(10220), pp.304-308.
The final message in the conclusion section has to be more current and aligned with ongoing planetary challenges and the impact of land -related issues (eg climate change) on health.
We have discussed land-related issues in terms of kaitiakitanga, or the care of the land by Māori as well as the role of the land in our cultural identity. This is highly relevant and extremely current to Māori as the care of the land that we live on is extremely important for the health of Māori and the sustainability of the land is our priority, and will continue to be.
Reviewer 3 Report
Really important project. Well written and clear. As a non-Maori reader, I appreciated the glossary at the end as it was helpful to reference throughout. I do wish I knew something about the authors and how they are positioned in relation to this project. That too is an important methodological consideration. Recommend for publication.
Author Response
Our team would like to extend our thanks to the reviewers for the time they took to review this paper. We appreciate their positive comments and valuable feedback. We purposely highlighted the links between Māori and the natural world as an intrinsic part of our cultural identity in a way that would be understandable for those who are not familiar with Māori culture. It is very important to us to preserve and maintain the cultural integrity of Rongoā Māori healing practices and we hope this article will highlight that.
Thank you for your comment about positioning ourselves as the authors in the paper. We have added a paragraph, starting on L48, to provide greater context about our perspective, as follows:
The authors of this article are all Māori. All are university educated; four are trained and practising researchers and one is a Rongoā practitioner, with many years of healing experience in the Rongoā sector. Individuals within the team are known within the Rongoā sector, either because of their previous research in the field, or due to their being a practitioner themselves. The team is passionate about maintaining the cultural integrity of Rongoā Māori, and this research was born out of a desire to promote a greater understanding of Rongoā. The fact that our traditional healing practices are misunderstood, or not known at all, by many in Aotearoa/New Zealand has, in part provided the impetus for this work.
There will also be an author bio next to the paper once it is published, which will provide more detail about ourselves as the authors, for readers as well.